# Fracture Resistance of New Metal-Free Materials Used for CAD-CAM Fabrication of Partial Posterior Restorations

**DOI:** 10.3390/medicina56030132

**Published:** 2020-03-18

**Authors:** Georgina García-Engra, Lucia Fernandez-Estevan, Javier Casas-Terrón, Antonio Fons-Font, Pablo Castelo-Baz, Rubén Agustín-Panadero, Juan Luis Román-Rodriguez

**Affiliations:** 1Prosthodontics and Occlusion Unit, Department of Stomatology, Faculty of Medicine and Dentistry, University of Valencia, 46021 Valencia, Spain; georgina.g.engra@gmail.com (G.G.-E.); javier.casas@uv.es (J.C.-T.); antonio.fons@uv.es (A.F.-F.); ruben.agustin@uv.es (R.A.-P.); juanluis.romanrodriguez@gmail.com (J.L.R.-R.); 2Co-director of Master’s Program in Endodontics, Restorative Dentistry, and Dental Esthetics, University of Santiago de Compostela, 15782 Santiago de Compostela, Spain; pablocastelobaz@hotmail.com

**Keywords:** fracture resistance, resin nanoceramic, polymer-infiltrated ceramic network, lithium disilicate, zirconium-reinforced lithium silicate

## Abstract

*Background and Objectives:* To evaluate in vitro the fracture resistance and fracture type of computer-aided design and computer-aided manufacturing (CAD-CAM) materials. *Materials and Methods:* Discs were fabricated (10 × 1.5 mm) from four test groups (*N* = 80; *N* = 20 per group): lithium disilicate (LDS) group (control group): IPS e.max CAD^®^; zirconium-reinforced lithium silicate (ZRLS) group: VITA SUPRINITY^®^; polymer-infiltrated ceramic networks (PICN) group: VITA ENAMIC^®^; resin nanoceramics (RNC) group: LAVA™ ULTIMATE. Each disc was cemented (following the manufacturers’ instructions) onto previously prepared molar dentin. Samples underwent until fracture using a Shimadzu^®^ test machine. The stress suffered by each material was calculated with the Hertzian model, and its behavior was analyzed using the Weibull modulus. Data were analyzed with ANOVA parametric statistical tests. *Results:* The LDS group obtained higher fracture resistance (4588.6 MPa), followed by the ZRLS group (4476.3 MPa) and PICN group (4014.2 MPa) without statistically significant differences (*p* < 0.05). Hybrid materials presented lower strength than ceramic materials, the RNC group obtaining the lowest values (3110 MPa) with significant difference (*p* < 0.001). Groups PICN and RNC showed greater occlusal wear on the restoration surface prior to star-shaped fracture on the surface, while other materials presented radial fracture patterns. *Conclusion:* The strength of CAD-CAM materials depended on their composition, lithium disilicate being stronger than hybrid materials.

## 1. Introduction

Partial coverage restorations make it possible to preserve an additional 20%–30% more dental structure than other more invasive restoration techniques, such as full coverage crowns [1]. The incidence of complications, both pulp (1.3% after 12.6 years) and periodontal, is lower with inlay, onlay, and overlay incrustations than complete coverage crowns [2,3,4]. The current survival rate of incrustations varies between 75 and 98% after 5 years, offering an effective therapeutic alternative to conventional techniques [5,6].

Computer-aided design and computer-aided manufacturing (CAD-CAM) technology has simplified the planning and fabrication procedures involved in partial coverage restorations and has also lead to the development of new materials with homogeneous structures that suffer less contraction when polymerized [7,8]. The development of metal-free and hybrid materials combining two main restoration components (resin and ceramic) has provided a wide range of materials with versatile indications and improved biomechanical properties. To date, CAD-CAM materials can be classified as ceramics (conventional feldspathic and high-strength ceramics) and hybrid materials [9,10,11]. High-strength CAD-CAM ceramics include IPS e.max CAD^®^ (Ivoclar Vivadent, Schaan, Liechtenstein), composed of lithium disilicate (LDS), and VITA SUPRINITY^®^ (VITA Zahnfabrik, Bad Zäckingen, Germany) catalogued as zirconium-reinforced lithium silicate (ZRLS), composed of two crystalline phases (lithium metasilicate and zirconium dioxide) [12,13,14]. Hybrid materials can be classified as polymer-infiltrated ceramic networks (PICN), such as VITA ENAMIC^®^ (VITA Zahnfabrik, BadZäckingen, Germany), and resin nanoceramics (RNC), such as LAVA™ ULTIMATE (3M, St. Paul, Minn, USA) [15,16,17] (Table 1) [18,19].

According to the literature, there is little consensus with regard to the biomechanical behavior of these new restoration materials. The increasing variety of materials with different compositions and physical properties require new research into their clinical behavior to increase understanding of how best to use them.

The objective of this study was to analyze the static compression resistance of three materials indicated for fabricating indirect restorations in the posterior region. The study’s null hypothesis (1) was that no significant differences would be found between the three materials tested in comparison with a control material. The second null hypothesis (2) was that hybrid materials would present higher fracture resistance than ceramic materials.

The present paper is the first of a two-part study; the second part investigates the same materials, focusing on the influence of immediate dentin sealing on their biomechanical behavior.

## 2. Materials and Methods

The study was conducted following directive ISO 6872:2015 for ceramic materials used in dental prosthetics [20]. The study design was approved by the University of Valencia Ethics Committee for Research Involving Human Subjects (Reg. No H1542128153508, approved 04.4.2019).

Discs were fabricated from the following materials using a Sirona InEos Blue^®^ and Inlab MC XL^®^ (Dentsply Sirona, York, PA, USA) to scan and milling machine: IPS e.max CAD^®^; VITA SUPRINITY^®^; VITA ENAMIC^®^; and LAVA™ ULTIMATE (*n* = 80, *n* per group = 20). Each disc had a 10 mm diameter and thickness of 1.5 mm—the thickness recommended by all the material manufacturers for partial coverage restorations in the posterior region [21,22,23]. All discs were polished, as recommended by the manufacturers for each one. Four groups were created according to the four materials tested (Table 1): LDS group (lithium disilicate—control group); ZRLS group (zirconium-reinforced lithium silicate); PICN group (polymer-infiltrated ceramic network); RNC group (resin nanoceramic) [12].

Eighty human molars were stored in physiological serum until preparation (no longer than 6 months after extraction). Their occlusal surfaces were prepared by a single clinician (G.G.-E.), cut 2 mm, to expose the surface of the dentine, and then was polished with medium and fine diamond burs (125–105 µm) and medium, fine, and extra-fine sof-lex™ discs (3M) [24]. Samples were checked under ultraviolet light (Sylvania S18W/BLB, Danvers, MA, USA) to ensure the occlusal enamel removal [25]. The prepared surfaces were etched with 37.5% orthophosphoric acid (enamel for 30 s and dentin for 15 s) and were washed and dried by means of negative suction [26,27,28]. Then, the adhesive agent corresponding to each material was applied and light-cured (Figure 1) (Table 2) [21,22,23].

The fitting surface of each disc was treated and cemented following the manufacturers’ instructions (Table 2). IPS e.max CAD^®^, VITA SUPRINITY^®^, and VITA ENAMIC^®^ were etched with 4.9% hydrofluoric acid (20 s for IPS e.max CAD^®^, and 60 s for VITA SUPRINITY^®^ and VITA ENAMIC^®^). Samples were washed and dried, and the silane coupling agent was applied for 1 min, and the corresponding adhesive was applied without light-curing. Discs of LAVA™ ULTIMATE were sandblasted with aluminum oxide particles (50 microns at 2 bar pressure) using a CoJet Prep^®^ (3M ESPE), washed, left to dry, and Scotchbond™ Universal (3M ESPE) single-step adhesive was applied without light-curing [21,22,23].

The discs were cemented onto the molar surfaces using the adhesives recommended for each material. Samples were light-cured for 5 s; excess cement was removed and was light-cured for a further 20 s. The disc-tooth complex was embedded in type IV plaster in a copper cylinder and was conserved in physiological serum for 24 h (Figure 1).

Each specimen underwent static compression testing until fracture with a Shimadzu AG-X Plus^®^ (Shimadzu corp., Kyoto, Japan) at a crosshead speed of 0.5 mm/min and a 100 KN load cell with an alumina ball (4 mm of diameter) (Figure 2) [29]. The fractured specimens were examined under a Leica M125^®^ optical microscope (Leica micro GmbH, Wetzlar, Germany) to determine each fracture pattern classified as an adhesive fracture (fracture of the disc-molar interface), cohesive fracture (internal structure discs’ fracture), and complete fracture (disintegration of the disc surface) [30,31]. The cohesive fracture was subdivided in indentation (deformation of the disc surface before fracture) and radial (star-shaped fracture of the disc surface). Statistical analysis was performed with the statistical software package SPSS for Windows version 20.0 (SPSS, Chicago, IL, USA). The data obtained were analyzed using ANOVA parametric tests: T2 multiple comparison tests of differences in fracture resistance between groups. Statistical significance was set at *p* < 0.05.

The probability of failure was estimated by means of the Weibull modulus and scale parameter (m) following the directive ISO 20501:2003/Cor 1:2009 for Weibull statistical analysis of fracture resistance data [32]. The confidence interval was 95%, and the power to detect differences between mean resistance values was 0.77.

## 3. Results

The mean fracture resistance values obtained in compression testing were as follows: group LDS, 4588.9 MPa ± 1843.5; group ZRLS, 4476.3 MPa ± 762.6; group PICN, 4014.2 ± 681.1, and group RNC, 3110.0 MPa ± 169.0.

Statistically significant differences between mean values were found; groups LDS (control group), ZRLS, and PICN showed significantly higher resistance to fracture than group RNC (*p* < 0.001, T2 de Tamhane test; *f* = 8.57). The ANOVA model did not identify significant differences between the LDS (control group), ZRLS, and PICN groups.

After calculating Weibull distribution, the largest distribution was obtained by the RNC group with a Weibull modulus of 16. The ZRLS and PICN groups presented similar plot gradients (m = 6.32 and 6.11, respectively). But the LDS control group obtained the smallest distribution (m = 2.36) (Figure 3) (Table 3).

Regarding fracture patterns (Table 4) (Figure 4), these varied between the materials analyzed. Glass ceramics underwent radial fractures to the disc surface without affecting the underlying substrate. Hybrid materials (groups PICN and RNC) presented indentation deformations of the surface before undergoing radial fractures.

## 4. Discussion

Lithium disilicate is considered an adequate material for fabricating metal-free ceramic partial restorations [33,34,35,36,37]. At the same time, the principles of biomimetics have guided the development of new hybrid materials that attempt to imitate the biomechanical behavior of natural dental tissue [38]. Factors influencing the success of partial restorations are various, the most important being the composition of the restoration material and the adhesion techniques employed [39]. In this context, it is necessary to carry out detailed investigations of all the structures involved and to understand their biomechanical properties.

The static load test is selected since is considered the best way to study the fracture resistance of brittle material. This mechanical property needs to use an axial force over the material. The use of a static load test or a flexural test (three points of load) is recommended to study this property. Although there are authors that lead to a fatigue test as the best way to obtain clinic results, Sieper [40] did not find statistical differences in the fracture resistance of CAD-CAM materials, with and without a cyclic load test, concluding that masticatory fatigue did not affect the fracture strength of crowns.

The present study investigated the behavior of three CAD-CAM materials—VITA SUPRINITY^®^, VITA ENAMIC^®^, and LAVA™ ULTIMATE, in comparison with IPS e.max CAD^®^, on teeth prepared to expose dentin. Analysis of the results rejected the first null hypothesis. The highest fracture resistance values were found in the control group of monolithic lithium disilicate ceramic specimens. The ZRLS and PICN groups (VITA SUPRINITY^®^ and VITA ENAMIC^®^, respectively) obtained lower fracture resistance values without significant differences in comparison with the control group. But the RNC group (LAVA™ ULTIMATE) obtained significantly lower fracture resistance, which rejected the second null hypothesis (2). Despite its lower elasticity modulus (Table 1)—and so more plastic behavior—its fracture resistance did not exceed the ceramic materials assayed.

The composition of both high-strength ceramics and polymer-reinforced ceramics includes a ceramic matrix that is sensitive to the action of hydrofluoric acid on its surface. This creates an effective bond at the material-cement interface [41,42]. But the composition of the resin nano ceramic LAVA™ ULTIMATE is acid-resistant, making it necessary to sandblast its surface to create micro retention [23,43]. The sandblasting process is affected by the clinician’s actions, and the duration of the process beyond 30 s or pressure higher than 0.2 MPa generates excess roughness and surface deterioration, which can compromise adhesion [10,44]. The procedure recommended by the manufacturer uses Scotchbond™ Universal adhesive. Its composition includes pre-hydrolyzed silane monomers, which are known to present less stability and efficacy than non-hydrolyzed silane [45]. These characteristics could influence the adhesion process of this resin nanoceramic, making its cement-restoration interface more susceptible to hydrolytic degradation in the oral medium in the long term [46].

The mean force exerted by the stomatognathic system ranges between 500 and 600 N [47]. Individuals with parafunctional masticatory habits apply much greater forces of between 900 and 1000 N [48]. In spite of the differences in strength obtained between the different materials assayed, all the CAD-CAM materials obtained sufficiently high values to resist the forces exerted by normal or parafunctional individuals. So, all the materials were adequate for partial restorations in the posterior sector, regardless of the presence of parafunctional habits.

An in-depth investigation of fragile materials, such as ceramic, will obtain a disperse data set corresponding to different stages and forms of development and different types of surface fracture [49]. The Weibull distribution analyzes the natural probability of a structure’s fracture mechanics, and so provides an indication of the material’s reliability that makes it possible to standardize the results of testing [50,51,52]. Analyzing the data generated in Weibull distribution, the control group (LDS group) obtained the lowest *m* value. In spite of presenting the highest fracture resistance, it was the material with the highest probability of accumulated failure in comparison with the other materials. The failure distribution in the ZRLS, PICN, and RNC groups was significantly larger than the control group, so these materials showed more predictable behavior and less probability of fracture (Figure 3).

Microscopy analysis of the hybrid materials observed a fracture pattern influenced by the inherent resilience of these materials. The aluminum ball used to test fracture resistance created deformation on the surfaces of both materials (PICN and RNC). But the glass-ceramics (LDS and ZRLS) did not present any surface deformation. Differences in occlusal surface wear depended on the material used for restoration. Hybrid materials, which contained organic components, presented more elastic behavior and so a lower Young’s modulus (Table 1). This characteristic might be beneficial as it offers greater protection of the antagonist’s teeth and so better conservation of dental structures [13]. In agreement with the present work, Lawson also reported the deformation of the surfaces of hybrid materials, affirming that RNC generates wear to its antagonist tooth that is similar to that produced by dental enamel. But PICN presents wear behavior that is more similar to feldspathic ceramics [17].

The studies reviewed in the present work report similar behaviors for the materials assayed. In a trial of minimal thickness restorations (0.5–0.8 mm), Al-Akhali [53] found higher strength with glass ceramics than resin matrix materials, the composition of the ceramic materials being the cause of their behavior. Sieper [40] obtained results showing the same tendency. In Sieper’s study, VITA SUPRINITY^®^ showed higher strength with a fracture resistance that was even higher for reduced thicknesses (0.8–1 mm) compared with IPS e.max CAD^®^. Likewise, in an assay of adhesion to dentin, Van den Breemer [46] found improvements in the behavior of ceramic materials when the adhesion technique involved immediate dentin sealing, as this established higher bond strength at the tooth-cement-restoration interfaces. However, Carvalho et al. [37] did not find significant differences when comparing the fracture resistance of lithium disilicate and RNC after cyclic isometric loading. Although they reported that the materials’ fracture resistance was inversely proportional to elasticity modulus, they found a small interval between the two materials, observing positive behavior and similar strengths.

To sum up, the fracture resistance of CAD-CAM restoration materials is influenced by both internal factors (composition, structure, thickness) and external factors (dental structure, dental surface exposure, adhesion technique, the cement employed, occlusion, etc.) [54,55,56]. The lack of consensus in the literature points to the need for in vitro studies to gain a better understanding of the biodynamic behavior of these materials.

The present study suffered a significant limitation in the fact that it tested fracture resistance using discs. Some authors have argued that the results obtained in this type of experiment might differ from clinical reality as the disc does not reproduce dental anatomy [57]. But others consider that this type of test does constitute an adequate method for assaying materials and has been seen to obtain homogeneous fracture resistance values [58]. We believe that the uniformity of the specimens and the load distribution to be essential elements in initial trials of materials in common usage.

## 5. Conclusions

Considering the limitations of this study, the following conclusions could be drawn:The CAD-CAM restoration materials analyzed showed high fracture resistance values that were adequate for use in partial coverage dental restorations in the posterior region.IPS e.max CAD^®^ ceramic obtained the highest fracture resistance, although Weibull distribution showed that it had less predictable behavior than the other materials tested.Hybrid materials presented lower fracture resistance than ceramic materials due to their internal composition.The resilient behavior exhibited by hybrid materials generated surface wear patterns prior to fracture, which implied more conservative behavior than ceramic materials.

## Figures and Tables

**Figure 1 medicina-56-00132-f001:**
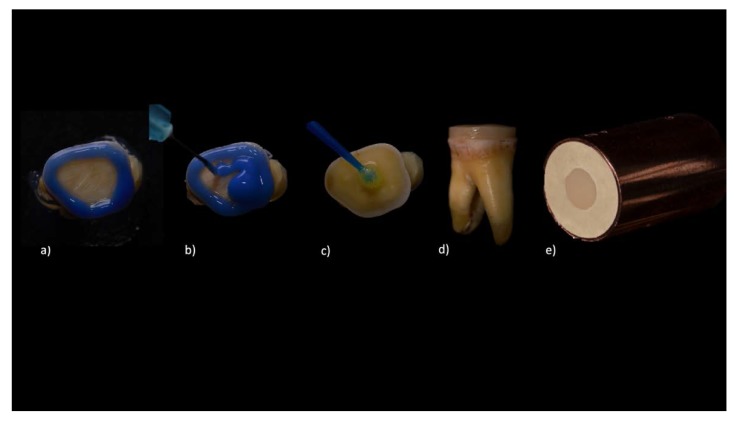
(**a**) Selective acid etching of peripheral enamel; (**b**) Dentin acid etching; (**c**) Adhesive application; (**d**) Disc cemented onto prepared molar; (**e**) Insertion in a copper cylinder.

**Figure 2 medicina-56-00132-f002:**
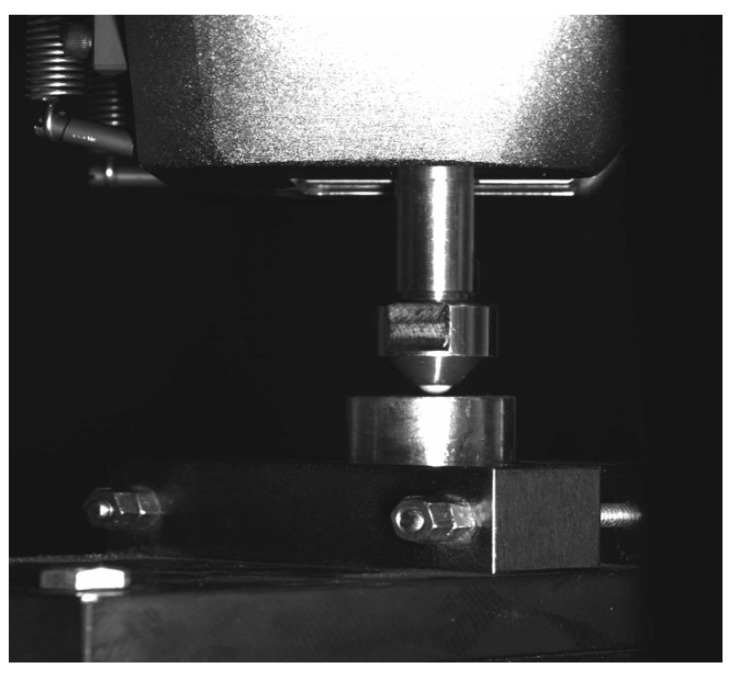
Sample tested in the Shimadzu AG-X Plus^®^ test machine.

**Figure 3 medicina-56-00132-f003:**
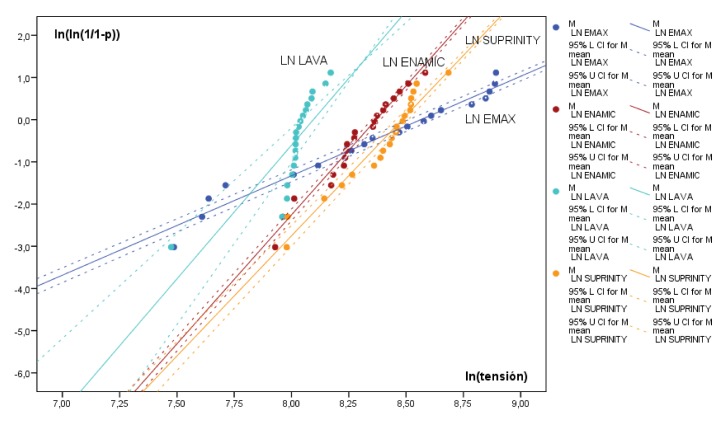
Graph showing Weibull distribution for groups LDS (lithium disilicate), ZRLS (zirconium-reinforced lithium silicate), PICN (polymer-infiltrated ceramic networks), and RNC (resin nanoceramics) and (upper and lower C.I. 95%) calculated.

**Figure 4 medicina-56-00132-f004:**
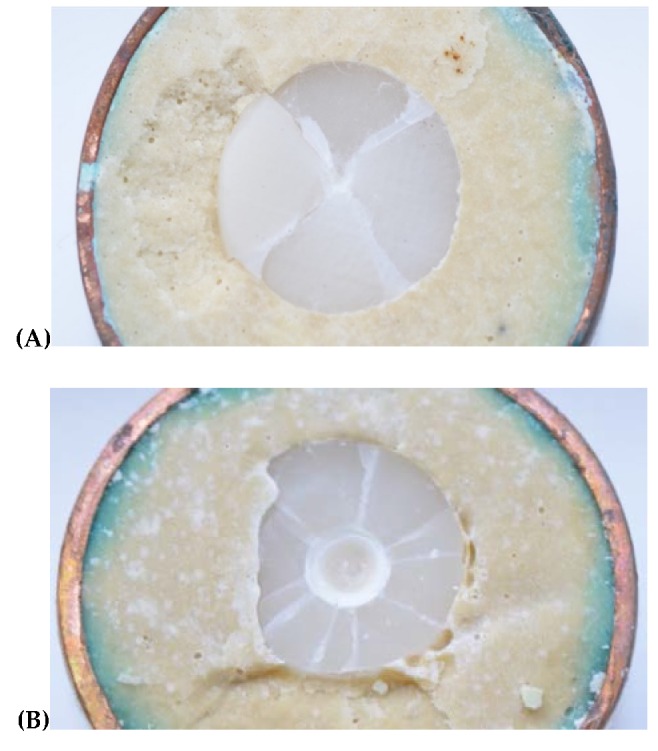
Samples fracture after compression testing. (**A**) Specimen of LDS group; (**B**) Specimen of RNC group.

**Table 1 medicina-56-00132-t001:** Study groups: Materials, composition (percentage in weight), and biomechanical properties.

Group	Material	Type	Manufacturing Data	Chemical Composition	Properties	Characteristics/ Lot nº
**LDS group (GC)**	IPS e.max CAD^®^	High-strength ceramic	Ivoclar Vivadent, Schaan, Liechtenstein	Crystalline phase: 70% lithium disilicate	Fracture resistance 360 MPa Elasticity modulus (95 GPa) Poisson modulus 0.25 Marginal fit 0.06 mm Acid-sensitive	U40015 U40016 MT BL4/C14
**ZRLS** **group**	VITA SUPRINITY^®^	High-strength ceramic	VITA Zahnfabrik, Bad Zäckingen, Germany	Crystalline phase: 64% lithium silicate 15% lithium disilicate 10% zirconium dioxide Glass-ceramic matrix	Fracture resistance 420 MPa Elasticity modulus (70 GPa) Poisson modulus 0.23 Marginal fit 0.06 mm Acid-sensitive	74740 74742 A2-HT PC-14
**PICN** **group**	VITA ENAMIC^®^	Hybrid material: PICN	VITA Zahnfabrik, Bad Zäckingen, Germany	Glass-ceramic matrix: 86% conventional feldspathic ceramic (leucite and zirconia) Organic phase: 14% UDMA and TEGDMA	Fracture resistance 160 MPa Elasticity modulus (30 GPa) Poisson modulus 0.23 Acid-sensitive	56171 82120 1M1-HT EMC-14
**RNC** **group**	LAVA™ ULTIMATE	Hybrid material: RNC	3M ESPE, St Paul, Minn, USA	Crystalline phase: 80% Nanoceramic (silica and zirconia) Organic matrix: 20% organic filling	Fracture resistance 250 MPa Elasticity modulus (12.77 GPa) Poisson modulus 0.30 Marginal fit 0.01 mm Acid-resistant	N429938 N429987 A3-LT/14

LDS, lithium disilicate; ZRLS, zirconium-reinforced lithium silicate; PICN, polymer-infiltrated ceramic networks; RNC, resin nanoceramics; CG, Control Group; UDMA, urethane dimethacrylate; TEGDMA, triethylene glycol dimethacrylate.

**Table 2 medicina-56-00132-t002:** Study groups, materials used in the cementation process, and procedures.

Group	Material and Cementation Procedure	Type	Chemical Composition	Duration	Manufacturer	Lot No.
**LDS group** **(CG)**	1. IPS Ceramic Etching Gel^®^	Acid etching for ceramic	4.9% hydrofluoric acid	20’	Ivoclar Vivadent	T76221
	2. IPS Ceramic Neutralizing Powder^®^	Neutralizing powder	Sodium carbonate 25%–50%, calcium carbonate 25%–50%.	20’	Ivoclar Vivadent	V47224
	3. Monobond Plus^®^	Silane	Adhesive monomers 4%, ethanol 96%	60’	Ivoclar Vivadent	X43365
	1. Excite DSC^® a^	Adhesive agent	Phosphonic acid acrylate, dimethacrylates, hydroxyethyl methacrylate, highly dispersed silicon dioxide, ethanol, catalysts, stabilizers, and fluoride	20’ rub 20’ light-cure	Ivoclar Vivadent	Z33289
	2. Variolink Esthetic DC Neutral^®^	Dual-cure resin cement	Barium glass filling, mixture of oxide 52.2%, dimethacrylate 22%, high dispersión silica, ytterbium trifluoride 25%, initiators and stabilizers 0.8%, pigments <0.1%	20’ light-cure each face	Ivoclar Vivadent	W95564 W95566
	3. Liquid Strip^®^	Glycerine	Glycerine gel	20’	Ivoclar Vivadent	K44713
**ZRLS group** **/** **PICN group**	1. Vita Ceramics Etch^®^	Acid etching for ceramic	5% hydrofluoric acid	60’	VITA Zahnfabrik	G32613
	2. VITASIL^®^	Silane	3-methacryloxypropyltrimethoxysilane, ethanol, and water	60’	VITA Zahnfabrik	I18532
	3. Vita A.R.T Bond^® b^	Adhesive agent	Bond: methacrylate 97%–99% and polyalkenoate 1%–3% Primer A: water 96%–98%, sodium fluoride <0.1%, organic substances 2%–4% Primer B: methacrylate 89%–91%, polyalkeonate 6%–8%, water 2%–4%	20’ primer 20’ light-cure adhesive	VITA Zahnfabrik	H15866
	4. Vita DUO CEMENT^®^	Resin cement dual	Methyl methacrylate 28%–32%, inorganic components 63%–77%	20’ light-cure each face	VITA Zahnfabrik	F72605
**RNC group**	1. Cojet Prep^®^	Sandblasting	Aluminium particles, particle size: 30 μm, pressure 2.0 bars	30’	3M ESPE	
	2. Scotchbond™ Universal Adhesive ^c^	Universal adhesive agent	BisGMA, HEMA, decamethylene dimethacrylate, ethanol, water, silane-treated silica, 2-propenoic acid, methacrylated phosphoric acid, copolymer of acrylic and itaconic acid, ethyl-4-dimethylaminobenzoat, camphorquinone, (dimethylamino) ethyl methacrylate, methyl ethyl ketone	20’ rub 20’ light-cure	3M ESPE	4636134
	3. RelyX ™ Ultimate	Dual-cure resin cement	Base paste: silane-treated glass powder, 2-propenoic acid, 2-methyl, reaction products with 2-hydroxy-1,3-propanedyl dimethacrylate and phosphorus oxide, TEGDMA, silane-treated silica, oxide glass chemicals, sodium persulfate, tertbutyl peroxy-3,5,5- trimethylhexanoate, copper acetate monohydrate Catalyst paste: silane-treated glass powder, substituted dimethacrylate, 1,12-dodecane dimethacrylate, silane-treated silica, 1-benzyl-5-phentyl-barbic-acid, calcium salt, sodium p-toluenesulfinate, 2-propenic acid, 2-methyl-, di-2,1-ethanediyl ester, calcium hydroxide, titanium dioxide	20’ light-cure each face	3M ESPE	4751433
**Tooth**	1. Scotchbond™ Universal Etchand	Acid etching agent Tooth	3 mL 37.5% orthophosphoric acid	15’ dentin 30’ enamel	3M ESPE	4638524
	2. Adhesive recommended for each material ^a,b,c^					

CG, Control Group; BisGMA, Bisphenol A-glycidyl methacrylate; HEMA, polymacon; TEGDMA, triethylene glycol dimethacrylate.

**Table 3 medicina-56-00132-t003:** Statistical analysis of fracture resistance data in MPa. Weibull Distribution for LDS, ZRLS, PICN, and RNC Groups. N: number of specimens; σ: mean fracture resistance (MPa); *m*: Weibull modulus; R2: the probability of failure.

	Groups			
	LDS Group (Control Group)	ZRLS Group	PICN Group	RNC Group
**N**	20	20	20	20
**σ (MPa)**	4588.6	4476.3	4014.2	3110.0
**σf ± SD**	1843.5	762.6	681.1	169.0
**95% CI of mean**	3725.8–5451.4	4119.4–4833.2	3695.5–4333.0	3030.9–3189.1
**Minimum (MPa)**	1784.1	2919.4	2775.7	2864.1
**Maximum (MPa)**	7277.9	5913.2	5344.3	3537.5
**Median (MPa)**	4508.2	4659.4	3925.1	3058.3
***m***	2.36	6.32	6.11	16
**R2**	0.97	0.942	0.971	0.636

**Table 4 medicina-56-00132-t004:** Distribution of numbers and types of fractures in each study group.

Fracture Type		LDS Group (CG)	ZRLS Group	PICN Group	RNC Group
**Adhesive**		0	0	0	0
**Cohesive**	**Indentation**	0	0	16 (80%)	15 (75%)
	**Radial**	19 (95%)	20 (100%)	16 (80%)	15 (75%)
**Complete**		1(5%)	0	4 (20%)	5 (25%)

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
