# Peer review of "Fracture Resistance of New Metal-Free Materials Used for CAD-CAM Fabrication of Partial Posterior Restorations"

_medicina, 2020, doi:10.3390/medicina56030132_

Round 1

Reviewer 1 Report

1. The manuscript is well written and presented. It is also very interesting and the figures are very professional. However, the rationale of the study is not completely understood. There are some previous studies regarding fracture resistance of metal-free polymer for dental use like; Rammelsberg P, et al. Fracture resistance of posterior metal-free polymer crowns. J Prosthet Dent 2000; 84: 303-8. What is the significance of this investigation compare to the previous studies? Please clarify

2. The manuscript should be revised throughout the paper to make in more readable. There are some errors in the manuscript. For example;

(1) “Discs were fabricated from the following materials using a Sirona InEos Blue ® and Inlab MC XL® (Dentsply Sirona, PA, U.S) to scan and milling machine: IPS e.max CAD®; Vita SUPRINITY®; VITA Enamic®; and Lava ULTIMATE (n=80, n group=20).”  n per group is more appropriate.

(2) But the LDS control group obtained the smallest distribution (m= 2.36) (Graph 1) (Table 3). What does the graph 1 mean? In the supplementary files, I found the graph 1. But I could not find the difference between Fig 1 and graph 1.

(3) In the discussion section, “Likewise, in an assay of adhesion to dentin, Van den Breemer found improvements in the behavior of ceramic materials when the adhesion technique involved immediate dentin sealing, as this established higher bond strength at the tooth-cement-restoration interfaces [48]. However, Carvalho et al. did not find significant differences when comparing the fracture resistance of lithium disilicate and RNC after cyclic isometric loading. Although they reported that the materials’ fracture resistance was inversely proportional to elasticity modulus, they found a small interval between the two materials, observing positive behavior and similar strengths [37].” Citations in text are usually added just after authors’ name. Please check the author guideline and revise.

(4) Also, in the reference part, some errors are found.

So, please clarify and revised the errors in the whole manuscript.

3. In discussion section (page 8), the authors stated the limitation of the study.

“The present study suffered a significant limitation in the fact that it tested fracture resistance using discs. Some authors have argued that the results obtained in this type of experiment may differ from the clinical reality as the disc does not reproduce dental anatomy [40]. But others consider that this type of test does constitute an adequate method for assaying materials and has been seen to obtain homogeneous fracture resistance values [41]. We believe that the uniformity of the specimens and the load distribution to be essential elements in initial trials of materials in common usage.”

I think it would be better that comments on the limitation of this study will be addressed just before the concluding paragraph for readability. So, please consider this paragraph moving backward

Author Response

REVIEWERS RESPONSE

  1. A) REVIEWER 1
  2. The manuscript is well written and presented. It is also very interesting and the figures are very professional. However, the rationale of the study is not completely understood. There are some previous studies regarding fracture resistance of metal-free polymer for dental use like; Rammelsberg P,et al.Fracture resistance of posterior metal-free polymer crowns. J Prosthet Dent 2000; 84: 303-8. What is the significance of this investigation compare to the previous studies? Please clarify

Reply: Thank you for your comment. We have reviewed the article suggested and we have incorporated in the discussion part and references.

In our study, new cad-cam materials are tested with the aim to know its mechanical behavior properties to minimally invasive posterior partial restorations. The new hybrid materials have a microstructure where ceramic and resin particles are combined. This characteristics let some advantages compared to direct resin restorations (greater fracture resistance, avoid shrinkage due to polymerizations, homogeneous internal structure,...) and compared to ceramic indirect restorations (less elastic module, repairability, less occlusal antagonist wear,..).  The results of our study show that hybrid materials have fracture resistance values higher than mastication levels, so that, are materials recommended to posterior partial restorations.

  1. The manuscript should be revised throughout the paper to make in more readable. There are some errors in the manuscript. For example;

(1) “Discs were fabricated from the following materials using a Sirona InEos Blue ® and Inlab MC XL® (Dentsply Sirona, PA, U.S) to scan and milling machine: IPS e.max CAD®; Vita SUPRINITY®; VITA Enamic®; and Lavaä ULTIMATE (n=80, n group=20).”  n per group is more appropriate.

Reply: Thank you for your comment. In the manuscript, it has been modified accordingly by your considerations.

(2) But the LDS control group obtained the smallest distribution (m= 2.36) (Graph 1) (Table 3). What does the graph 1 mean? In the supplementary files, I found the graph 1. But I could not find the difference between Fig 1 and graph 1.

Reply: Thank you for your comment. There was a mistake on the name of the graph 1. All the supplementary files have been modified. Graph 1 represents the Weibull distribution of the four tested groups. The dots are the neperian logarithm representations of each sample tested, the continuous line represents the trend line of each group and the upper and lower dot line represent de confidence interval at 95% of each group.  Control group, with the lower Weibull Modulous, has the lower slope in the graphic 1 in comparison with the other groups, which means that is the less reliable materials.

(3) In the discussion section, “Likewise, in an assay of adhesion to dentin, Van den Breemer found improvements in the behavior of ceramic materials when the adhesion technique involved immediate dentin sealing, as this established higher bond strength at the tooth-cement-restoration interfaces [48]. However, Carvalho et al. did not find significant differences when comparing the fracture resistance of lithium disilicate and RNC after cyclic isometric loading. Although they reported that the materials’ fracture resistance was inversely proportional to elasticity modulus, they found a small interval between the two materials, observing positive behavior and similar strengths [37].” Citations in text are usually added just after authors’ name. Please check the author guideline and revise.

Reply: Thank you for your comment.  In the manuscript, it has been modified accordingly.

(4) Also, in the reference part, some errors are found.

So, please clarify and revised the errors in the whole manuscript.

Reply: Thank you for your comment. We have reviewed all the references and adapted them appropriately to the format of the magazine.

  1. In discussion section (page 8), the authors stated the limitation of the study.

“The present study suffered a significant limitation in the fact that it tested fracture resistance using discs. Some authors have argued that the results obtained in this type of experiment may differ from the clinical reality as the disc does not reproduce dental anatomy [40]. But others consider that this type of test does constitute an adequate method for assaying materials and has been seen to obtain homogeneous fracture resistance values [41]. We believe that the uniformity of the specimens and the load distribution to be essential elements in initial trials of materials in common usage.”

I think it would be better that comments on the limitation of this study will be addressed just before the concluding paragraph for readability. So, please consider this paragraph moving backward

Reply: Thank you for your comment.  In the manuscript, it has been moved accordingly.

Reviewer 2 Report

The authors conducted a well designed in-vitro study about certain mechanical properties of different conventional and hybride ceramics. The results are clearly presented and discussed. Thus, the manuscript should be of interest for the journal's readers and could be published in its present form.

Author Response

REVIEWERS RESPONSE

B) REVIEWER 2

The authors conducted a well designed in-vitro study about certain mechanical properties of different conventional and hybride ceramics. The results are clearly presented and discussed. Thus, the manuscript should be of interest for the journal's readers and could be published in its present form.

Reply: Thank you for your comment. 
